# Effect of TRPV4 Antagonist GSK2798745 on Chlorine Gas-Induced Acute Lung Injury in a Swine Model

**DOI:** 10.3390/ijms25073949

**Published:** 2024-04-02

**Authors:** Meghan S. Vermillion, Nathan Saari, Mathieu Bray, Andrew M. Nelson, Robert L. Bullard, Karin Rudolph, Andrew P. Gigliotti, Jeffrey Brendler, Jacob Jantzi, Philip J. Kuehl, Jacob D. McDonald, Mark E. Burgert, Waylon Weber, Scott Sucoloski, David J. Behm

**Affiliations:** 1Lovelace Biomedical Research Institute, Albuquerque, NM 87108, USA; nsaari@lovelacebiomedical.org (N.S.); andrewm.nelson@hmh-cdi.org (A.M.N.); rlbulla@sandia.gov (R.L.B.); krudolph@lovelacebiomedical.org (K.R.); agigliotti@lovelacebiomedical.org (A.P.G.); jbrendler@lovelacebiomedical.org (J.B.); jjantzi@lovelacebiomedical.org (J.J.); pkuehl@lovelacebiomedical.org (P.J.K.); jake.m@envolbio.com (J.D.M.); wweber@ara.com (W.W.); 2GSK, Collegeville, PA 19426, USA; mathieu.x.bray@gsk.com (M.B.); scott.sucoloski@gsk.com (S.S.); david.j.behm@gsk.com (D.J.B.)

**Keywords:** transient receptor potential vanilloid 4, pulmonary edema, mechanical ventilation, acute respiratory distress syndrome (ARDS), acute lung injury (ALI)

## Abstract

As a regulator of alveolo-capillary barrier integrity, Transient Receptor Potential Vanilloid 4 (TRPV4) antagonism represents a promising strategy for reducing pulmonary edema secondary to chemical inhalation. In an experimental model of acute lung injury induced by exposure of anesthetized swine to chlorine gas by mechanical ventilation, the dose-dependent effects of TRPV4 inhibitor GSK2798745 were evaluated. Pulmonary function and oxygenation were measured hourly; airway responsiveness, wet-to-dry lung weight ratios, airway inflammation, and histopathology were assessed 24 h post-exposure. Exposure to 240 parts per million (ppm) chlorine gas for ≥50 min resulted in acute lung injury characterized by sustained changes in the ratio of partial pressure of oxygen in arterial blood to the fraction of inspiratory oxygen concentration (PaO_2_/FiO_2_), oxygenation index, peak inspiratory pressure, dynamic lung compliance, and respiratory system resistance over 24 h. Chlorine exposure also heightened airway response to methacholine and increased wet-to-dry lung weight ratios at 24 h. Following 55-min chlorine gas exposure, GSK2798745 marginally improved PaO_2_/FiO_2_, but did not impact lung function, airway responsiveness, wet-to-dry lung weight ratios, airway inflammation, or histopathology. In summary, in this swine model of chlorine gas-induced acute lung injury, GSK2798745 did not demonstrate a clinically relevant improvement of key disease endpoints.

## 1. Introduction

Chlorine stands as one of the most extensively manufactured and utilized chemicals worldwide. Accidental encounters with chlorine gas are common occurrences in both occupational and domestic environments. Moreover, documented instances reveal substantial public exposures resulting from auto/rail accidents linked to the transportation of chlorine. Chlorine gas is also classified as an important chemical warfare agent due to the potential to cause mass casualties from its acute respiratory effects. Exposure to chlorine gas results in dose-dependent respiratory signs, which can include acute lung injury, acute respiratory distress syndrome, respiratory failure, and death in severe cases [1]. To date, there are no mechanistic-based therapeutic strategies for chlorine-induced acute lung injury. The current recommended response measures are supportive and aimed at the management of symptoms. Humidified oxygen is provided as a standard of care, based on the patient’s oxygen status. Beta-2-agonists (e.g., albuterol) in combination with ipratropium bromide are standard therapies to mitigate bronchospasm and airway irritation [2]. Inhaled or systemic corticosteroids are administered to decrease airway inflammation and lung edema, though quantification of the clinical benefit is difficult because they are typically administered with other treatments. Nebulized sodium bicoarbonate is a supplemental therapy that has been shown to provide some additional clinical benefit based on pulmonary function testing and quality of life assessments. Other experimental treatments, including sodium nitrate, dimethylthiourea, and rilopram, have demonstrated significant efficacy in preclinical chlorine injury models, but have not been evaluated in the clinic [3].

Although the pathophysiology of chlorine inhalation toxicity is poorly defined, acute lung injury is hypothesized to result in part from alveolar damage and barrier disruption secondary to formation of reactive oxygen and nitrogen species following reaction of chlorine with the airway mucosa [1]. Transient Receptor Potential Vanilloid 4 (TRPV4) is a non-selective calcium permeable cation channel that has been identified as an important regulator of alveolo–capillary barrier integrity in the lung [4,5,6]. Antagonism of this ion channel has been shown to be protective in mouse models of ventilator-induced lung injury [7,8], and more recently, TRPV4 has been proposed as a therapeutic target for COVID-19-induced pulmonary edema in humans [9]. In the context of chlorine inhalation toxicity, both genetic and pharmacologic inhibition of TRPV4 reduced pulmonary edema and inflammation and improved pulmonary function and oxygen saturation in chlorine-exposed mice [10]. 

GSK2798745 is a selective inhibitor of TRPV4 shown to be well-tolerated in several early clinical trials [11,12,13,14,15]. Whether GSK2798745 may be effective as a treatment of noncardiogenic pulmonary edema secondary to acute lung injury from chemical inhalation (e.g., an unexpected chlorine exposure in a potential mass casualty situation) has yet to be reported. The objective of this study was to evaluate the effectiveness of GSK2798745 as a post-exposure therapeutic in a mechanically ventilated Yorkshire Swine model of chlorine gas-induced acute lung injury. Here, we describe the results of two studies aimed at (1) optimizing and characterizing the lung injury phenotype following inhalation exposure to increasing durations of chlorine gas; and (2) evaluating the dose-response effectiveness of the intravenous GSK2798745 TRPV4 inhibitor, administered as a post-exposure therapeutic, in the optimized animal model. 

## 2. Results

Two independent studies were conducted in Yorkshire Swine: (1) chlorine inhalation model development study (*n* = 24) and (2) GSK2798745 dose-response efficacy study (*n* = 63). 

### 2.1. Chlorine Gas Exposure Resulted in Acute Dose-Dependent Pulmonary Functional Changes over 24 h

The goal of the chlorine inhalation model development study was to identify the optimal conditions that resulted in reproducible lung injury without lethality. The target injury was characterized by a reduction in the ratio of partial pressure of oxygen in arterial blood to the fraction of inspiratory oxygen concentration (PaO_2_/FiO_2_; P/F) to between 200–300, sustained over 24 h, satisfying the diagnostic criteria for acute lung injury in humans [16]. Response to inhalation of 240 parts per million (ppm) chlorine gas was assessed across a range of exposure durations, from 19 min to 90 min (*n* = 1–6/group, Appendix A). 

At exposure durations below 60 min (*n* = 14), survival was 100% through 24 h post-exposure. Survival was reduced to 62.5%, however, at exposure durations between 60–90 min (*n* = 8). The baseline P/F of swine ranged from 333 to 485, with an average baseline P/F of 390 (±9 SEM), across both air and chlorine-exposed animals. The immediate reduction in P/F following chlorine exposure averaged 42% (i.e., from 390 to 228) compared with baseline (Figure 1a) and was not correlated with exposure duration (Figure 1b). During chlorine exposure, lung resistance increased in a time-dependent manner to >25 cmH_2_O/L/sec in every animal, while dynamic compliance (C_dyn_) decreased to <5 mL/cmH_2_O (Appendix A).

The area under the curve (AUC) of the P/F recorded between hours 20–24 (P/F_20–24_) was evaluated across exposure durations to determine the optimal exposure conditions and reproducibility across animals. In general, there was an exposure duration-dependent effect on the P/F_20–24_, with lower P/F_20–24_ observed following longer exposures (Figure 2a). In animals exposed to chlorine for 45 min or less, P/F was not reliably sustained below 300 (Appendix A). In animals exposed to chlorine for 50 min or more, P/F was sustained below 300, but there was reduced survival with exposures of 60 min or greater (Appendix A; Appendix A). This led to the selection of the 55-min exposure of 240 ppm chlorine as the optimal conditions for the target injury. 

Other measures of functional lung injury—oxygenation index (OI), peak pressure (P_peak_), and C_dyn_—were also measured across exposure durations to determine the dose-dependence and variability in response for each of these phenotypic endpoints measured as AUC between hours 20–24 (Figure 2b–d). Target thresholds of interest included P_peak_ of at least 25 cmH_2_O, and C_dyn_ of at most 15 mL/cmH_2_O, on average. The increases in OI (Figure 2b) and P_peak_ (Figure 2c) with chlorine exposure were somewhat variable, and an exposure duration-dependent relationship was not statistically significant for these endpoints. C_dyn_ was more dramatically decreased with chlorine exposure in this model system, with values below 15 mL/cmH_2_O in animals exposed to chlorine for 45 min or greater on average (Figure 2d).

### 2.2. Chlorine Inhalation Resulted in Dose-Dependent Changes in Airway Responsiveness, Inflammation and Edema

Airway responsiveness was assessed by inhaled methacholine (MCh) challenge performed approximately 24 h post-chlorine exposure. Response to MCh was measured by the provocative MCh concentration required to increase lung resistance by 200% (i.e., PC_200_). This was enhanced with chlorine exposure in a dose-dependent manner for animals exposed for up to 50 min. At exposure times of 55 min or longer, however, PC_200_ values were not correlated with chlorine dose (Figure 3). This is likely owing to higher starting lung resistance values (measured 24 h post-exposure) with greater chlorine exposure levels (Appendix A); thus, the magnitude of increase in lung resistance required to achieve a doubling effect in response to MCh was greater with longer chlorine exposure times, and in some cases may have been physiologically impossible. 

Wet-to-dry lung weight ratios were calculated as a measure of pulmonary edema. A linear fit through these ratio values by exposure time, with air-exposed animals representing an exposure of 0, demonstrated a statistically significant (*p* = 0.003) positive slope for wet/dry lung weight ratio by exposure time, suggesting a chlorine dose-dependent increase in pulmonary edema (Figure 4), consistent with published models of chlorine inhalation injury in rodents [17,18], rabbits [19], and swine [20]. Chlorine exposure also resulted in a dose-dependent shift in airway inflammation, toward increasing percentages of neutrophils (Figure 5a) and decreasing percentages of macrophages (Figure 5b), based on four-parameter logistic curve fits to the data by exposure time. These results mirror pulmonary inflammatory skewing following chlorine exposure in mice and rats [17,18]. 

Chlorine-exposed animals showcased indications of airway injury and inflammation, distributed in a gradient across the respiratory tract. The trachea and proximal bronchi exhibited the most severe pathology, featuring extensive ulceration, fibrin accumulation, neutrophilic inflammation, reduced bronchial epithelial cell presence, and the absence of cilia. In the lower airways, intermittent sloughed epithelia formed sheets, blocking smaller bronchioles and alveolar ducts, whereas the alveolar epithelia remained largely unaffected (Figure 6). These observed changes in tissue structure closely parallel the outcomes documented in swine subjected to 100–140 ppm chlorine through mechanical ventilation [20]. 

### 2.3. GSK2798745 Treatment Resulted in Marginal Improvement of PaO_2_/FiO_2_ Ratios following Chlorine Exposure

Based on the pre-determined target threshold of P/F ratio between 200–300, representing mild acute respiratory distress syndrome in humans, a 55-min chlorine exposure was selected for the follow-on efficacy study to evaluate GSK2798745 dose response. Animals were randomly assigned to receive either vehicle or one of five doses of GSK2798745 (0.0006 to 5.94 mg/kg, with approximate 10-fold escalation intervals, *n* = 6–16 per group, Appendix A), delivered as an intravenous infusion within 15 min of exposure. Plasma GSK2798745 pharmacokinetics scaled approximately linearly with dose, and the maximum concentration was measured within 0.25 h of the end of the infusion (Appendix A). The circulating levels of GSK2798745 obtained with the three highest doses (0.0714 mg/kg, 0.792 mg/kg and 5.94 mg/kg) were anticipated to fully inhibit TRPV4 in the pig based on modelling predictions which take into account the plasma concentration of GSK2798745 necessary to fully inhibit lung edema in an TRPV4 agonist-induced rat model [21], and considering differences in PK, plasma protein binding and TRPV4 potency for GSK2798745 between pigs and rats. Submaximal inhibition was anticipated at the two lowest doses (0.0006 mg/kg and 0.0072 mg/kg). 

When comparing the AUC of the P/F ratios from hours 20–24 (P/F_20–24_), partial normalization was observed post-exposure in GSK2798745-treated animals. When fitting a four-parameter dose-response curve to P/F_20–24_, the estimated difference between the maximum and minimum parameters was 16.7 units (95% CI: (0.601, 32.7)). As the bounds of the 95% confidence interval for the range were above zero, there was evidence of a non-zero improvement in P/F_20–24_ with escalating dose. Despite marginal dose-dependent improvement, P/F_20–24_ for each of the GSK2798745 treatment groups remained below 300 (Figure 7a). Partial, but not significant, normalization of oxygenation index relative to vehicle were observed with higher doses of GSK2798745 (Figure 7b). Evaluating P/F ratio results at each hourly measurement over the course of the post-exposure monitoring period, we observe significant reduction in P/F ratio in the vehicle dose group, relative to air controls, across all but one hour (Hour 17). However, we observe only transient, inconsistent differences in P/F ratio in other dose groups relative to vehicle (Figure 8). Mild improvements in oxygenation were not accompanied by any measurable effects of GSK2798745 on mechanical lung function, including P_peak_, P_mean_, P_plat_, C_dyn_ or R_L_ measurements as an AUC between 20 and 24 h post-exposure (Appendix A). 

### 2.4. GSK2798745 Treatment Did Not Impact Airway Responsiveness, Pulmonary Edema or Airway Inflammation following Chlorine Exposure

In the dose-response study, airway response to inhaled MCh challenge was not heightened in chlorine-exposed animals, as measured by PC_200_ values based on change in R_L_, nor were there any detectable differences in airway response with GSK2798745 treatment (Figure 9a). As suggested by the model development data (Figure 3), it is possible that the increase in lung resistance that follows a 55-min chlorine exposure is large enough that subtle differences in airway response to MCh cannot be detected. Thus, evaluation of airway responsiveness with this endpoint may not be useful in this model. 

Compared with animals exposed to room air, animals exposed to chlorine demonstrated greater wet-to-dry lung weight ratios (1.18-fold; *p* = 0.035). Treatment with GSK2798745, however, did not improve wet/dry lung weight ratios compared with vehicle, regardless of dose (Figure 9b). Total protein measured from bronchoalveolar lavage fluid collected at the time of necropsy was used as an additional measure of vascular leakage. Compared with animals exposed to room air, animals exposed to chlorine demonstrated significantly greater bronchoalveolar lavage fluid (BALF) total protein (5.87-fold; *p* < 0.0001), but no change was observed with GSK2798745 treatment, regardless of dose (Figure 9c). 

Airway inflammation was assessed at the 24 h timepoint by total cell counts and cell differentials measured in BALF. Airway inflammation was driven primarily by recruitment of neutrophils and macrophages, which were both significantly increased in chlorine-exposed, vehicle-treated animals as compared with air controls. Total cell counts remained unchanged, however, with GSK2798745 treatment (Figure 9d,e). Similarly, microscopic evaluation of respiratory tissue revealed a similar magnitude of inflammation and injury in both the trachea and lung across all chlorine-exposed treatment groups (Figure 9f,g), suggesting that GSK2798745 had no detectable effect on these endpoints. 

Collectively, these data indicate that the mild improvement in P/F ratio observed with GSK2798745 treatment was not associated with any significant effects on respiratory mechanics, airway inflammation, pulmonary edema, or histopathology, as could be evaluated within the context of this study.

## 3. Discussion

The contributions of the studies presented herein are twofold, and independent data are provided to support (1) the characterization of a new chlorine inhalation swine model following exposure to 240 ppm chlorine for 19 to 90 min; and (2) a dose-response efficacy evaluation of the candidate TRPV4 antagonist, GSK2798745, against chlorine gas-induced acute lung injury. Establishment of a clinically relevant model of chlorine inhalation acute lung injury has been challenging based on the variability of clinical disease observed in humans depending on exposure dose, duration, and underlying disease that can all impact clinical outcome [1]. Several models of chlorine gas inhalation in swine have been reported in both spontaneous-breathing and ventilated animals [20,22,23,24,25,26,27]. Chlorine gas results in immediate changes in breathing patterns during exposure, so the inhaled dose and resulting disease phenotype is highly variable in spontaneous-breathing animals [26]. Thus, a ventilated delivery model was employed for these studies to control the target chlorine dose based on exposure duration. In other published ventilated swine models, responses to 140 ppm chlorine for 10 min [20,22] and to 400 ppm chlorine for 15–20 min [23,24,25] have been characterized through 5–23 h post-exposure. The character and kinetics of lung injury observed in this model were similar to those reported following a 15 min exposure to 400 ppm chlorine [24], though the magnitude of injury varied for certain endpoints. The injury produced following at least 50-min exposure to 240 ppm chlorine was triphasic, characterized by an initial rapid impairment of gas exchange and derangement of mechanical lung function. This was followed by a gradual improvement in functional and mechanical measurements over the course 6–12 h, and relatively sustained lung injury thereafter through 24 h post-exposure. The clinical phenotype was relatively consistent and can be described as a mild to moderate acute lung injury based on P/F ratios between 200–300 [28], dynamic compliance below 15 mL/cmH_2_O, and peak inspiratory pressure sustained above 25 cmH_2_O at 20–24 h post-exposure. Moreover, a 55-min exposure to 240 ppm chlorine did not result in any mortality with the protocols employed for these studies. To our knowledge, this is the first report in swine to describe phenotypic changes as a function of chlorine dose that was controlled by exposure duration versus chlorine concentration. Here, we describe that the predicted changes in airway response to inhaled methacholine (MCh) challenge, wet-to-dry lung weight ratios, and relative populations of airway inflammatory cells were all modulated in a chlorine exposure-duration dependent manner. 

Using the 55-min/240 ppm chlorine exposure model as a platform, the efficacy of intravenous treatment with GSK2798745 was evaluated. Based on a dose-response curve-fit, a 16.7 unit increase in P/F ratio (as measured at hours 20–24) was observed with GSK2798745 from 0 to 5.94 mg/kg, suggesting non-zero effect with escalating treatment. This was accompanied by mild, but non-significant, improvements in oxygenation index in animals treated with higher doses of GSK2798745. There were no treatment-related effects in mechanical lung function, airway response to MCh, wet-to-dry lung weights, airway inflammation or histopathology endpoints. Despite some evidence of improvement in oxygenation following treatment with GSK2798745, the magnitude of these changes was not sufficient to suggest translation to a clinically meaningful effect. In addition, GSK2798745 failed to impact multiple key secondary endpoints, and the collective data do not support TRPV4 antagonism via GSK2798745 as an effective treatment strategy for chlorine-induced acute lung injury in this swine model.

A significant challenge in this study arose from the necessity to choose a specific chlorine dose and dosing protocol to serve as a representative disease model for assessing the efficacy of GSK2798745. This limitation stems from the inherent variability in the disease phenotype resulting from chlorine exposure in humans, which is influenced by factors such as dose, exposure duration, and underlying health conditions. Thus, it is impossible to capture all the possible clinical scenarios with a single animal model. A robust efficacy study, however, necessitates a model with a well-defined and reproducible phenotype in order to detect meaningful changes in key disease endpoints. The 55 min/240 ppm ventilated chlorine exposure paradigm used in this study resulted in increased wet-to-dry lung weight ratios and increased total protein within lung lavage fluid as evidence of pulmonary edema and alveolo–capillary barrier impairment, which were hypothesized to be mitigated with GSK2798745 treatment. Although fluid dysregulation in the lung contributes to reduced oxygen tension, the magnitude of associated changes was overall mild within 24 h of chlorine exposure, and therefore not likely to be a major contributor to observed impairments of gas exchange during the acute disease. Consistent with the acute presentation following chlorine inhalation in humans [29], histopathology revealed evidence of lower airway obstruction from desquamation of upper airway epithelia, suggesting obstructive airway disease as another significant component of the functional respiratory impairment in this model. Treatment with GSK2798745 would not have been expected to mitigate the concurrent effects of obstructive airway disease, which may have masked more subtle effects on alveolo–capillary barrier integrity and pulmonary fluid regulation. It is possible, therefore, that the available therapeutic window for this endpoint may have been too small for detection of efficacy in this model. A more severe or more chronic injury model characterized by more substantial pulmonary edema may reveal greater therapeutic potential of GSK2798745 in the context of chlorine inhalation.

There were several additional limitations to this study. First, although GSK2798745 lacked profound efficacy when dosed after chlorine exposure, it is possible that dosing GSK2798745 prior to, during, or immediately following chlorine exposure could have prevented chlorine injury. Although this potential exists, the aim of this study was to assess GSK2798745 as a medical countermeasure to an unexpected chlorine exposure in a potential mass casualty situation. In this study, GSK2798745 treatment was initiated 15 min post termination of chlorine exposure, approximating the minimal amount of time a first responder could arrive and administer a medical countermeasure. This differs from a prior chlorine exposure study in mice where TRPV4 antagonists were administered immediately after chlorine exposure and attenuated lung injury [10]. It is unclear whether these disparate results between studies are due to differences in inhibitor dosing timings relative to chlorine treatment, species, or another factor. Irrespective of differences in TRPV4 antagonist dosing time relative to chlorine treatment, it is also possible that the mouse is a more translationally relevant model for assessing chlorine countermeasures than swine. However, this is not likely considering the anatomical and physiological similarities between swine and human lungs and the general acceptance of swine as a large animal model for translational respiratory research [30]. Interestingly, consistent with the lack of profound GSK2798745 efficacy in this swine chlorine exposure model, in humans, GSK2798745 failed to attenuate segmental lipopolysaccharide (LPS)-induced elevation of bronchoalveolar lavage (BAL) total protein or neutrophils, despite blood and lung exposures that were predicted to be efficacious [14]. Although the lung injury agents were different (chlorine versus LPS), both resulted in elevated BAL neutrophils which were not altered by GSK2798745, suggesting the swine model is translationally relevant. Finally, although it is possible the lack of profound GSK2798745 efficacy in this swine model was due to inappropriate pharmacology (e.g., inadequate compound levels at the site of action), this is not likely the case. Based on modeling predictions which consider in vivo efficacy of GSK2798745 in a rat lung edema model and corrections for cross-species plasma protein and TRPV4 potency, GSK2798745 was anticipated to be fully efficacious in this swine model at circulating concentrations achieved at the top three doses (0.0714 mg/kg, 0.792 mg/kg and 5.94 mg/kg GSK2798745) [11,21]. In addition, although not directly measured in this swine study, BAL GSK2798745 levels in humans were 3.0- to 8.7-fold higher than those in plasma, suggesting that insufficient GSK2798745 levels in BAL were not likely the cause for lack of efficacy in this swine study [14].

## 4. Conclusions

In summary, exposure of swine to 240 ppm chlorine gas for over 50 min resulted in an acute lung injury characterized by significant and sustained changes in lung function over 24 h, which was accompanied by characteristic inflammation and pathology. Treatment with GSK2798745 at pharmacologically relevant doses in a 55-min exposure model, however, did not result in a significant improvement of these endpoints. Based on these data, it is unlikely that TRPV4 inhibition with GSK2798745 will be efficacious as a stand-alone treatment for chlorine-induced acute lung injury. 

## 5. Materials and Methods

### 5.1. Study Designs

Two independent studies were conducted on Yorkshire Swine: (1) chlorine inhalation model development study (*n* = 24) and (2) GSK2798745 dose response efficacy study (*n* = 63). Individual animal demographics and exposure/treatment assignments are detailed in Appendix A. All animal studies were conducted in accordance with the GSK policy on the care, welfare, and treatment of laboratory animals under protocols reviewed and approved by the Lovelace Biomedical Research Institute animal care and use committee. All evaluations were approximately 24 h, and animals were fully anesthetized and mechanically ventilated for the duration of the study.

### 5.2. Animals

Yorkshire Swine were sourced from Premier BioSource (Ramona, CA, USA). Animals ranged from 11–36 weeks of age and 25–55 kg at the time of enrollment on study. Prior to shipment, animals were tested and confirmed negative for Brucella abortus, Mycoplasma hyopneumoniae, Leptospira spp., Transmissible Gastroenteritis Virus, Pseudorabies Virus, Influenza A Virus, Porcine Reproductive and Respiratory Syndrome Virus, and Porcine Epidemic Diarrhea Virus. Prior to enrollment on study, animals were examined by a clinical veterinarian, and clinical pathology (hematology and serum chemistry) was evaluated. Only healthy animals were enrolled in the study. Animals were acclimated for a minimum of one week prior to study start. Whenever possible, animals were socially housed with sex-matched pen mates. Animals were fed Envigo Teklad 8753C Miniswine diet (Envigo, Madison, WI, USA), and received water ad libitum. Animals were fasted overnight prior to sedation for anesthesia. All animal studies were conducted in accordance with the GSK policy on the care, welfare, and treatment of laboratory animals under protocols reviewed and approved by the Lovelace Biomedical Research Institute animal care and use committee, and in compliance with all applicable sections of The Animal Welfare Act Regulations (9 CFR Parts 1, 2, 3) and the Guide for the Care and Use of Laboratory Animals. Animal facilities were fully accredited by the Association for Assessment and Accreditation of Laboratory Animal Care (AAALAC) International.

### 5.3. Randomization and Blinding

Chlorine exposures were performed on 1–2 animals per day. Due to the staggered nature of the model development and efficacy studies, animals were received in cohorts of 12–20 animals at a time. Upon arrival, each cohort of pigs was weighed, and these body weights were used to determine the order of enrollment in the study, with the heaviest animals of each sex enrolled first, and the lightest animals of each sex enrolled last within that cohort. This enrollment strategy was intended to minimize weight differences between animals at the time of exposure, as swine of this age range gain weight rapidly, resulting in a significant increase in body weight from the time of arrival to the time of exposure for the last animals within a study cohort. The treatment schedule for the GSK2798745 efficacy study was randomized, and with the exception of the technicians who formulated the test article, all study personnel were blinded to test article treatment for animals exposed to chlorine. In-life study personnel were not blinded to control animals exposed to room air. Personnel were unblinded to the treatment groups following all data compilation. The study pathologist was temporarily unblinded to slides from *n* = 3 animals from the air control group, chlorine-exposed vehicle-treated group, and from the high dose treatment group in order to establish a range of injury to be used for the subsequent blinded histologic scoring. The slides from the *n* = 9 animals that were temporarily unblinded were then re-blinded and placed randomly into the full set of slides for blinded scoring.

### 5.4. Animal Preparation, Anesthesia and Mechanical Ventilation

Animals were sedated with an intramuscular injection of Telazol^®^ and maintained under anesthesia with a constant rate infusion of midazolam, fentanyl, propofol, and dexmedetomidine, plus lactated Ringer’s solution. Animals were intubated with a cuffed endotracheal tube. Except during chlorine exposure, mechanical ventilation was provided with an AVEA CareFusion ventilator using a volume-controlled mode, with tidal volume maintained at approximately 10–12 mL/kg, respiratory rate at 8–20 breaths per minute, inspiratory to expiratory ratio of approximately 1:2, and a peak end expiratory pressure (PEEP) of 5 cmH_2_O. The fraction of inspired oxygen (FiO_2_) was maintained at 21% (i.e., no supplemental oxygen) during all pre-chlorine exposure activities. Central venous and arterial catheters were placed in either femoral or cervical vessels under ultrasound guidance, and urinary Foley balloon catheters were placed surgically using a suprapubic approach. Following surgery, animals were placed in the prone position and were allowed to equilibrate for at least 15 min prior to obtaining baseline pulmonary function measurements. Vital parameters—including heart rate, body temperature and peripheral oxygen saturation (SpO_2_)—were measured once every 15 min throughout the duration of the study. Following chlorine exposure, if SpO_2_ fell below 90% for more than 15 min and concurrently measured PaO_2_ was <60 mmHg, the FiO_2_ was increased incrementally above 21% until SpO_2_ was greater than or equal to 90%.

### 5.5. Chlorine Exposures

Tanks of chlorine gas (240 parts per million balanced in medical air) were sourced from Airgas, Inc. Chlorine was fed directly to a Newport Medical HT-70 ventilator equipped with Teflon-lined breathing circuits for delivery to the animal. A schematic of the chlorine exposure system is shown in Appendix A. Animals were placed in a prone position for chlorine (or room air) exposure. Chlorine (or air) was delivered at a tidal volume of 12 mL/kg and a respiratory rate of 20 breaths per minute with PEEP set to 5 cmH_2_O. For animals enrolled in the model development study, chlorine exposure duration ranged from 19 to 90 min and air exposures were 60 min. For animals enrolled in the efficacy study, both the chlorine and air exposure duration were 55 min. 

### 5.6. Chlorine Dose Calculations

The targeted and actual chlorine doses (mg) for each animal were calculated based on chlorine concentration (i.e., 240 ppm or 0.57 mg/L) and either the targeted or actual ventilation volume (L), as determined from the ventilator or pneumotachometer, respectively, using the following formulas (Appendix A): 

Targeted Cl_2_ Dose (mg/kg)
$$=\frac{0.57\frac{mg\;Cl_{2}}{L}\times TV\;\left(L\right)\times RR\left(bpm\right)\times Exposure\;Duration\;\left(min\right)}{Body\;Weight\;(kg)},$$
where TV = tidal volume setting on ventilator and RR = respiratory rate setting on ventilator.

Actual Cl_2_ Dose (mg/kg)
$$=\frac{0.57\frac{mg\;Cl_{2}}{L}\times Total\;Accumulated\;Volume\;\left(L\right)\;}{Body\;Weight\;(kg)},$$
where the total accumulated volume is calculated by the pneumotachometer over the course of the entire exposure.

### 5.7. GSK2798745 Formulation, Administration and Bioanalysis

GSK2798745 was formulated as a solution in 20% Captisol^®^ (*w*/*v*) and 3% dimethyl sulfoxide (*v*/*v*) in water. The concentration of the formulation was adjusted such that each treatment group was dosed with the same total volume based on body weight (i.e., 6 mL/kg). Animals were treated with either vehicle or with one of five doses of GSK2798745 ranging from 0.0006 to 5.94 mg/kg, with approximate 10-fold escalation intervals (*n* = 6–16 per group), delivered as a 90-min intravenous infusion beginning within 15 min of the end of chlorine (or air) exposure. Dose formulation analyses of GSK2798745 formulations were performed by liquid chromatography tandem mass spectrometry (LC-MS/MS) to confirm dose. Briefly, analytical samples were diluted with water and acetonitrile containing the internal standard and mixed well by vortexing. The samples were analyzed with a Sciex API 5000 using a reverse phase gradient on a Waters Acquity UPLC HSS T3 column. Fexofenadine was used as the internal standard. 

Plasma was collected at baseline and 0.25, 0.5, 1, 2, 3, 4, 6, 8, 12, and 24 h following the end of the GSK2798745 infusions for bioanalysis. Plasma GSK2798745 concentrations were measured by LC-MS/MS using LBRI Methods BSM-1249 and BSM-1269. Briefly, plasma samples (K_3_EDTA) were extracted by protein precipitation with acetonitrile containing the internal standard. The samples were analyzed with a Sciex API 5000 using a reverse phase gradient (10 mM ammonium bicarbonate in water and acetonitrile) on a Waters Acquity UPLC HSS T3 column. Fexofenadine was used as the internal standard. 

### 5.8. Pulmonary Function Measurements

During chlorine (or room air) exposure, pulmonary function was continuously monitored from a pneumotachometer placed at the end of the endotracheal tube and esophageal catheter, advanced to the mid-thorax, both of which were coupled to differential pressure transducers. The pressure and flow signals were phase-matched and integrated using EMKA iox 2.10.5.28 software to calculate lung resistance (R_L_) and dynamic lung compliance (C_dyn_). Additional baseline and hourly post-exposure respiratory function measurements—including peak inspiratory pressure (P_peak_), mean airway pressure (P_mean_), dynamic lung compliance (C_dyn_), airway plateau pressure (P_plat_), and respiratory system resistance (R_L_)—were recorded directly from the AVEA ventilator. PaO_2_ measured from arterial blood analyzed with an iSTAT 1 Point of Care Wireless Analyzer (Abbott) was used to calculate PaO_2_/FiO_2_ (P/F) ratios. The concurrent mean airway pressure measurements were used to calculate the oxygenation index (OI) using the following formula:$$OI=\frac{FiO_{2}\times P_{mean}}{PaO_{2}}$$

### 5.9. Methacholine Challenge

Approximately 24 h post-exposure, animals were challenged with inhaled methacholine (MCh) to evaluate airway hyperresponsiveness. Escalating doses of beta-methacholine chloride were aerosolized using a vibrating mesh nebulizer placed in line with the breathing circuit. Changes in lung resistance and dynamic lung compliance were measured following each dose using methods described above. 

### 5.10. Euthanasia and Terminal Tissue Collections

Following completion of the MCh challenge test, animals were euthanized by intravenous injection of euthanasia solution (Euthasol) for gross necropsy and terminal tissue collections. Wet-to-dry lung ratios were determined from the right cranial and right caudal lung lobes. The right middle lung lobe was lavaged with 20 mL sterile saline for quantification of total cell counts, cell differentials, and total protein. The caudal trachea and left lung were instilled with 10% neutral buffered formalin. Formalin-fixed, paraffin-embedded sections were mounted on glass slides and stained with Hematoxylin and Eosin. Microscopic examination was performed by a board-certified veterinary pathologist. Lung tissues were examined for epithelial cell degeneration/sloughing, mixed cell inflammation, pulmonary edema, atelectasis, fibrosis, and presence of increased alveolar macrophages. The trachea was examined for epithelial cell degeneration/soughing and neutrophilic inflammation. The incidence and severity of each lesion was scored on a scale of 0–5 (0 = absent, 1 = minimal, 2 = mild, 3 = moderate, 4 = marked, 5 = severe).

### 5.11. Cytology

Lungs were removed at necropsy and the right middle lung lobe was separated from remaining lung. The lobe was instilled in the major airway with 10 mL of phosphate buffered saline and the available fluid immediately withdrawn. This process was performed one more time, and these 2 lavagates were pooled. Lavagates were then centrifuged at 1000× *g* for 10 min, and the cell pellet resuspended in saline. Cytospin slides were prepared and stained with a modified Wright–Giemsa stain and differential counts conducted by a trained Medical Laboratory Technologist (ASCP). An aliquot of the suspension was added to an equal volume of neutral buffered formalin with acridine orange (1:1000) solution for nuclear staining. Cell nuclei were then counted using the fluorescent channel on an automated cell counter (Nexcelom Cellometer^®^ Vision), and the cell number value multiplied by the differential percentage to obtain absolute values for various cell types.

### 5.12. Statistical Methods

Analyses were conducted in R 4.1.1, with packages ‘emmeans’ (1.7.0) for ANOVA post-hoc comparisons and contrasts, ‘drc’ (3.0.1) for dose-response (i.e., logistic) curve fitting, ‘pscl’ (1.5.5) for zero-inflated binomial model fitting, ‘AER’ (1.2.9) for Tobit-model fitting, and ‘PMCMRplus’ (1.9.3) for Kruskal–Wallis testing with Dunn’s post-test. Adjusted *p*-values of 0.05 or lower were considered statistically significant. 

In this study, there are two separate datasets containing the same endpoints (i.e., R_L_ and C_dyn_)—one dataset generated using the EMKA system during chlorine exposure, and one dataset generated from the AVEA ventilators following exposure. These two datasets were analyzed independently, and results generated from one instrument cannot be compared with those generated from the other, as the instruments were not calibrated to each other. 

To evaluate the pulmonary function, as measured by AUC of PaO_2_/FiO_2_, OI, P_peak_, and C_dyn_ across hours 20–24, as well as MCh challenge results and wet/dry lung ratio during model development, linear models were fit to each endpoint versus exposure time, with air-exposed animals representing an exposure of 0. Note that for MCh challenge, at exposure times of 55 min or greater, PC_200_ values were not correlated with chlorine exposure. To evaluate inflammatory profiles during model development, four-parameter logistic curves were fit to percentage data for polymorphonuclear neutrophils (PMNs) and macrophages versus exposure time.

Separately, smooth curve fits, representing the mean trajectory across animals for pulmonary functional changes observed in real-time during chlorine exposure in model development, were obtained using generalized additive models (via the ‘mgcv’ (1.8.36) package in R). Generalized additive model fits were also used to measure changes in oxygenation status measured over 24 h during model development, with trajectories and 95% confidence ribbons presented separately for air controls, chlorine exposures of less than 30 min, chlorine exposures between 40 and 45 min, and chlorine exposures 50 min and above (Appendix A). 

To evaluate efficacy of escalating treatment, one-way ANOVA was conducted on study endpoints, with post-hoc comparisons between results from each experimental group versus vehicle (with Dunnett’s adjustment for multiple comparisons). Endpoints were log-transformed prior to analysis where appropriate to stabilize variance in situations with high right-skew. For analyses of pulmonary function from post-exposure monitoring (PaO_2_/FiO_2_, OI, C_dyn_, R_L_, P_plat_, P_mean_, P_peak_), analysis was conducted on the AUC for results obtained between hours 20–24 post-exposure. Focusing on a time range closer to the end of study provides adequate time for the compound to distribute and impact pathophysiology; utilizing the full AUC (0–24 h) dilutes the potential to observe efficacy. We chose the 20–24 h timepoints, as opposed to the single 24 h timepoint, to reduce the variability of data collected at a single timepoint. Note that calculated AUC was divided by the time taken to bring the values back to scale, akin to an average.

Additional analyses were conducted on PaO_2_/FiO_2_, a primary endpoint for pulmonary function. First, efficacy was evaluated by fitting a four-parameter dose-response curve to the AUC results from chlorine-exposed animals (air controls excluded), and extracting the estimated range (i.e., the difference between the “maximum” and “minimum” parameters) and 95% confidence bounds. The pre-specified clinical success criterion for this study was identified as an increase of at least 25 units in PaO_2_/FiO_2_. For this method, the effect would have been deemed statistically significant if the bounds of the 95% confidence interval for the range of the dose-response was entirely above zero, providing evidence (at the alpha = 0.05 level) that the effect observed was non-zero. Second, the hourly results across the entire post-exposure monitoring period for P/F ratio were evaluated using a repeated measures mixed-modeling approach to compare the dose groups at each time point, while accounting for correlation between successive readings from the same individual animals. Statistical contrasts were calculated comparing each dose group (as well as air controls) relative to vehicle, with Dunnett’s adjustment applied at each hour to correct for multiple comparisons.

For the BALF endpoints, the absolute counts of PMNs, lymphocytes, macrophages, and mast cells were calculated, and analysis was conducted using zero-inflated negative binomial models, to account for the distributions of the endpoints as counts, as well as the high proportion of zeros in the results. Analysis was not conducted on eosinophils: ~81% of responses for eosinophils were either zero or missing. For BALF total protein, as well as MCh challenge results, analyses were conducted on the log-transformed outcomes, using a Tobit model to account for the presence of observations not evaluated or not quantifiable past a maximum assessed level (i.e., right-censoring). The upper limit for total protein was 200 mg/dL. For MCh challenge, PC_200_ results were adjusted by the resistance at hour 24. The maximum assessed level was 10 mg/mL MCh in the model development study, and 100 mg/mL MCh in the GSK2798745 dose-response study.

For histology results, scores for evaluations on the trachea and lung were summed and evaluated using a Kruskal–Wallis test for differences in distribution between groups, with Dunn’s post-test to compare experimental groups versus vehicle. 

## Figures and Tables

**Figure 1 ijms-25-03949-f001:**
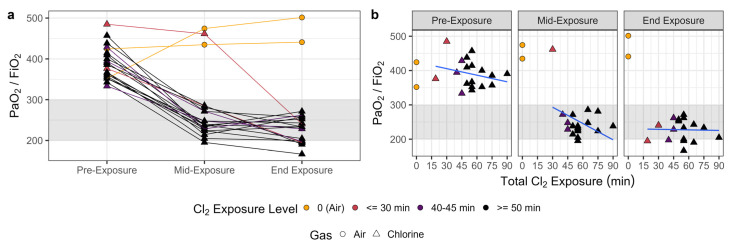
Chlorine inhalation exposure resulted in dose-dependent changes in PaO_2_/FiO_2_ (P/F) ratios. Swine were exposed to 240 ppm chlorine gas (Cl_2_, *n* = 19; triangles) or room air (*n* = 2; circles) via controlled ventilation for between 19 and 90 min. Arterial PaO_2_ was measured at baseline, midway through exposure, and immediately post-exposure, depending on their specific exposure duration. The P/F ratio was calculated assuming a 21% FiO_2_. Trends within individual animals (**a**) and correlation of P/F with exposure duration (**b**) are shown. Results are presented for air controls (in yellow), chlorine exposures of less than 30 min (in red), chlorine exposures between 40 and 45 min (in purple), and exposures 50 min and above (in black). The data in (**b**) are presented with linear fits (in blue) through the data points excluding air controls.

**Figure 2 ijms-25-03949-f002:**
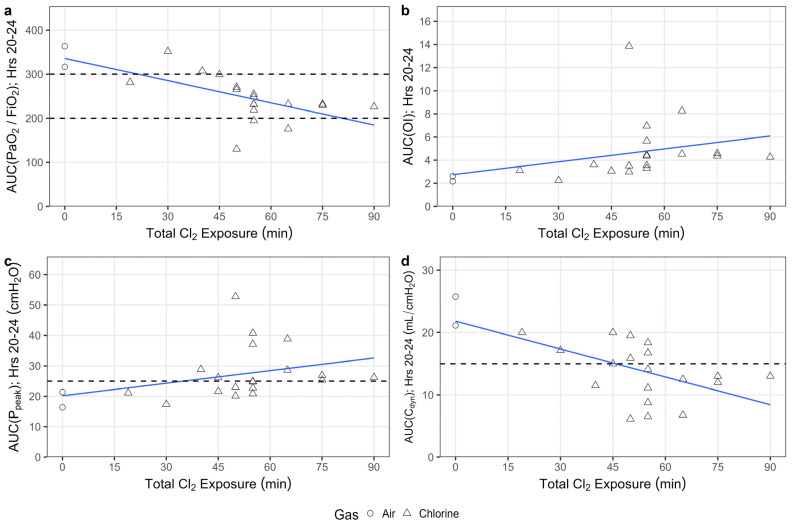
Chlorine inhalation exposure resulted in functional lung injury at 20–24 h post-exposure. Swine were exposed to 240 ppm chlorine gas (Cl_2_, *n* = 19) or room air (*n* = 2) via controlled ventilation for between 19 and 90 min. The PaO_2_/FiO_2_ (P/F) ratio (**a**), oxygenation index (OI) (**b**), peak airway pressure (P_peak_) (**c**) and dynamic compliance (C_dyn_) (**d**) were recorded once hourly following the end of exposure, with the area under the curve (AUC) over the last 4 h (i.e., 20–24) represented above. Dashed lines represent the target thresholds for P/F between 200–300, for P_peak_ of at least 25 cmH_2_O, and C_dyn_ of at most 15 mL/cmH_2_O. Results are presented with linear fits (in blue) through the data points, with air-exposed animals representing an exposure of 0.

**Figure 3 ijms-25-03949-f003:**
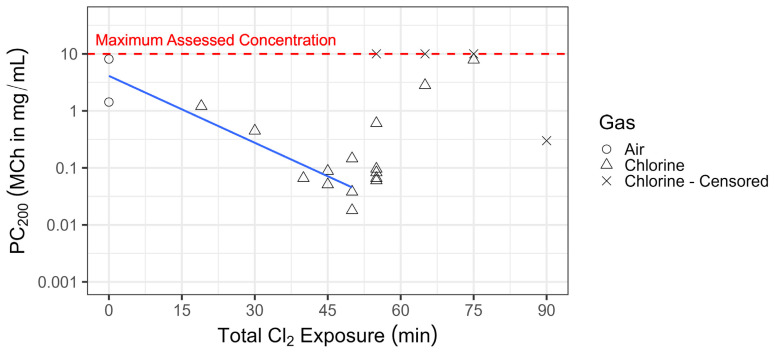
Chlorine inhalation exposure resulted in dose-dependent changes in airway responsiveness. Swine were exposed to 240 ppm chlorine gas (Cl_2_, *n* = 22) or room air (*n* = 2) via controlled ventilation for between 19 and 90 min. In surviving animals, at approximately 24 h post-exposure, lung resistance was measured following inhaled methacholine (MCh) challenge, and the provocative MCh concentration resulting in a 200% increase in lung resistance (R_L_)(PC_200_) was calculated. Animals for which R_L_ failed to increase by 200% at the maximum MCh concentration (10 mg/kg) were censored. The data are presented with a linear fit (in blue) through the data points up to 50 min, with air-exposed animals representing an exposure of 0; at exposure times of 55 min or greater, PC_200_ values were not correlated with chlorine exposure.

**Figure 4 ijms-25-03949-f004:**
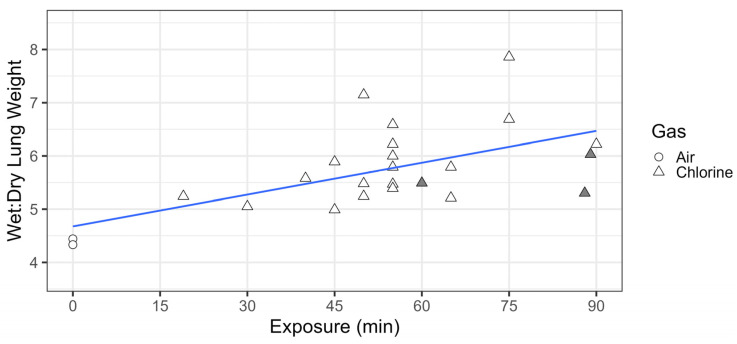
Chlorine inhalation exposure resulted in dose-dependent changes in pulmonary edema. Swine were exposed to 240 ppm chlorine gas (Cl_2_, *n* = 22) or room air (*n* = 2) via controlled ventilation for between 19 and 90 min. Wet-to-dry lung weight ratios were calculated from animals euthanized at approximately 24 h post-exposure (unfilled data points), or when they met moribund criteria (shaded data points). The data are presented with a linear fit (in blue) through the data points, with air-exposed animals representing an exposure of 0.

**Figure 5 ijms-25-03949-f005:**
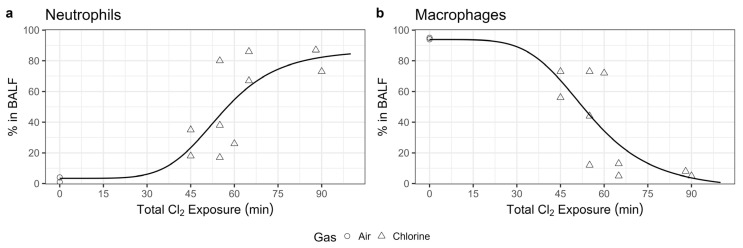
Chlorine inhalation exposure resulted in dose-dependent changes in airway inflammation. Swine were exposed to 240 ppm chlorine gas (Cl_2_, *n* = 22) or room air (*n* = 2) via controlled ventilation for between 19 and 90 min. Bronchoalveolar lavage fluid was collected from a subset of animals for total cell counts and cell differentials. The data for relative neutrophil (**a**) and macrophage (**b**) populations are presented with a four-parameter logistic curve fit through the data points.

**Figure 6 ijms-25-03949-f006:**
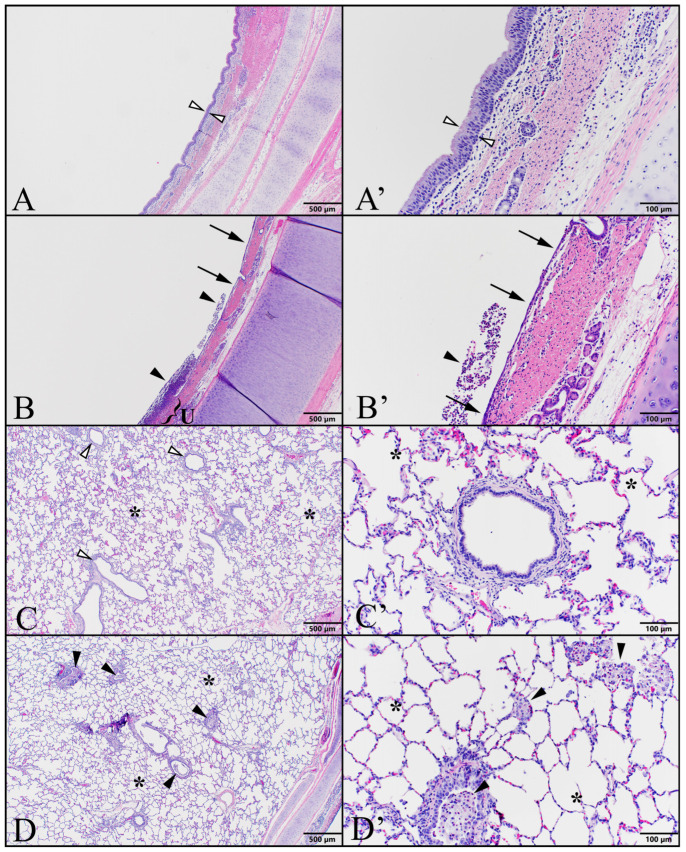
Chlorine inhalation exposure resulted in a gradient distribution of pathology along the respiratory tract. Swine were exposed to 240 ppm chlorine gas via controlled ventilation for between 19 and 90 min, and animals were euthanized at 24 h for necropsy and histopathology of respiratory tract tissues. Illustrative photomicrographs of trachea (series **A**,**B**) and lung (series **C**,**D**) are shown from an unexposed control (#7440; series **A**,**C**) and from an animal exposed for 19 min (#7443; series **B**,**D**). Low magnification (left column **A**–**D** via 4× objective; bar = 500 µm) and higher magnification (right column **A’**–**D’** via 20× objective; bar = 100 µm) images are shown. (**A**,**A’**) Trachea, Control: images demonstrate unremarkable lining of ciliated respiratory epithelium (open arrowheads). (**B**,**B’**) Trachea, Chlorine-exposed: inflammatory cell accumulation (arrowheads) is present within the tracheal lumen, with an underlying area of ulceration (U) demonstrating adherent inflammatory cells and fibrin. Remaining degenerate epithelium (arrows) is markedly thinner than in controls. (**C**,**C’**) Lung, Control: small/lower airways (open arrowheads) are empty allowing unrestricted airflow to normal gas exchange areas of alveolar parenchyma (*). (**D**,**D’**) Lung, Chlorine-exposed: small/lower airways often contain debris composed of sloughed epithelium and inflammatory cells (arrowheads). Alveolar parenchyma (*) is largely within normal limits.

**Figure 7 ijms-25-03949-f007:**
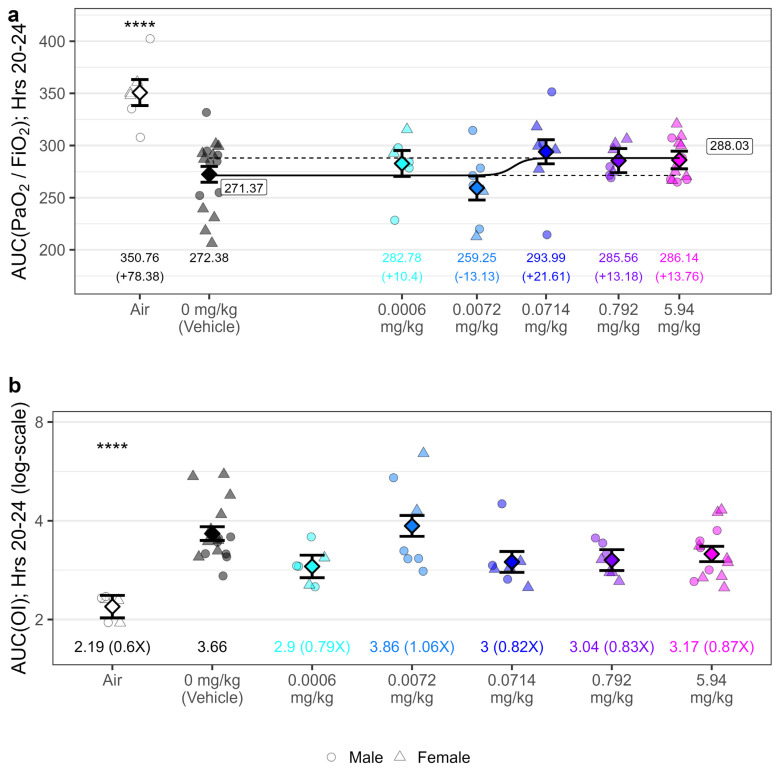
GSK2798745 treatment was associated with marginal improvement of PaO_2_/FiO_2_ but not oxygenation indices following chlorine exposure. Yorkshire Swine were exposed to either 240 ppm chlorine gas or room air by intratracheal inhalation for 55 min. Following exposure, animals were treated with either vehicle or GSK2798745 by intravenous infusion (*n* = 6–16 per treatment group). PaO_2_/FiO_2_ (**a**) and oxygenation index (OI, (**b**)) were calculated hourly post-exposure for 24 h, and the area under the curve (AUC) for measurements from 20–24 h post-exposure is presented. Individual PaO_2_/FiO_2_ and oxygenation index data points (jittered horizontally to remove overlap) are displayed for each treatment group, overlayed with 4-parameter dose-response model fit to chlorine-exposed swine data for PaO_2_/FiO_2_, with the levels and difference compared to vehicle control listed underneath (as calculated based on an ANOVA fit to data). Responses for air controls were not included in analysis of dose-response for PaO_2_/FiO_2_. Dashed lines represent minimum and maximum dose-response PaO_2_/FiO_2_ ratios estimated from model, with values labelled. Between-treatment comparisons to vehicle: **** = *p* < 0.0001.

**Figure 8 ijms-25-03949-f008:**
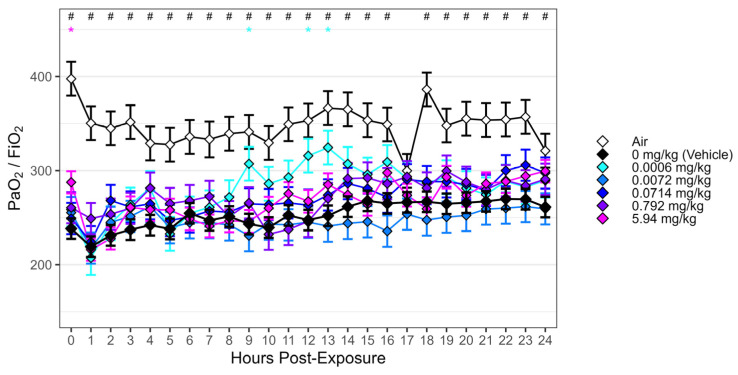
Chlorine exposure was associated with reduction in PaO_2_/FiO_2_ compared to air controls, but GSK2798745 treatment displayed only transient, inconsistent differences relative to vehicle. Yorkshire Swine were exposed to either 240 ppm chlorine gas or room air by intratracheal inhalation for 55 min. Following exposure, animals were treated with either vehicle or GSK2798745 by intravenous infusion (*n* = 6–16 per treatment group). PaO_2_/FiO_2_ was calculated hourly post-exposure for 24 h. Mixed-effects model estimates (diamonds) and standard errors are displayed for each dose group at each hour. Between-treatment comparisons to vehicle (Dunnett’s adjustment at each hour): # = *p* < 0.05 (versus air); * = *p* < 0.05 (versus GSK2798745 dose groups).

**Figure 9 ijms-25-03949-f009:**
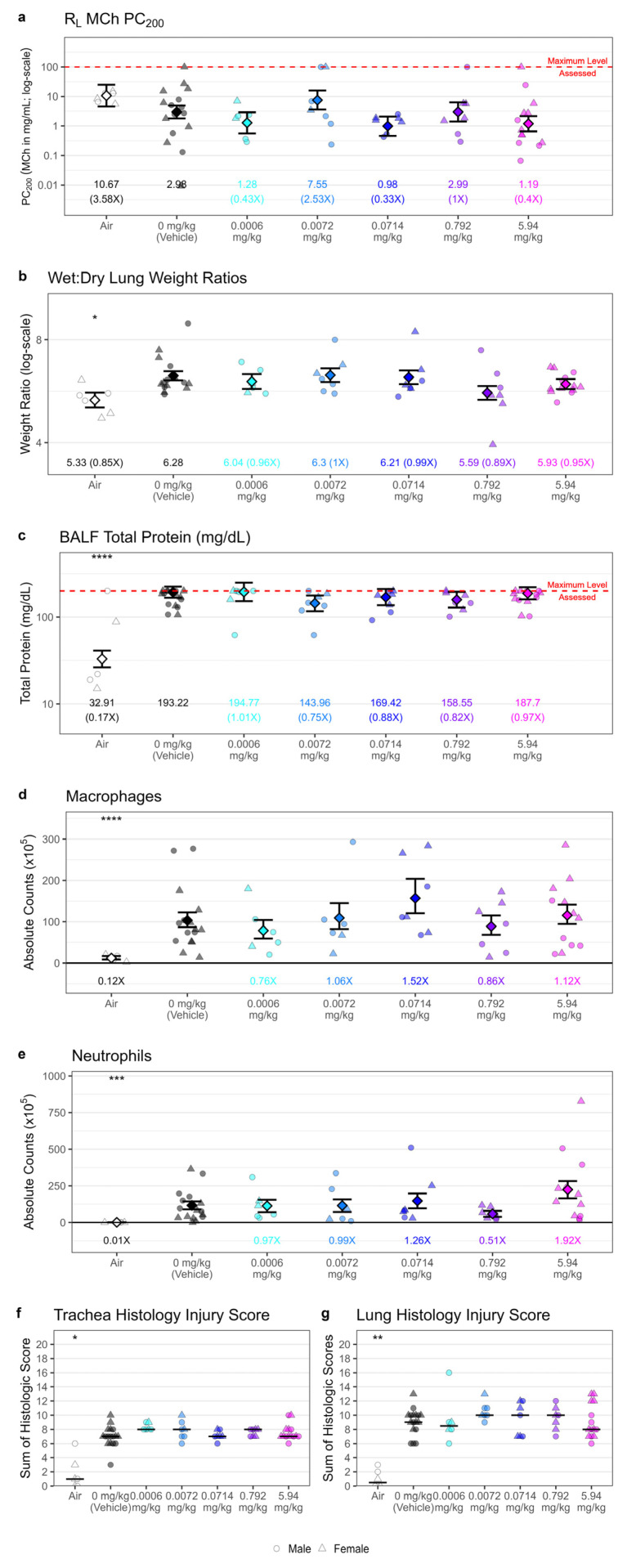
GSK2798745 treatment did not impact airway responsiveness, pulmonary edema, airway inflammation, or histopathology. Yorkshire Swine were exposed to either 240 ppm chlorine gas or room air by intratracheal inhalation for 55 min. Following exposure, animals were treated with either vehicle or GSK2798745 by intravenous infusion (*n* = 6–16 per treatment group). Approximately 24 h post-exposure, lung resistance (R_L_) was measured following inhaled methacholine (MCh) challenge, and the provocative MCh concentration resulting in a 200% increase in R_L_ (PC_200_) was calculated (**a**). Wet-to-dry lung weight ratios were calculated and averaged from the right cranial and right caudal lung lobes (**b**). Bronchoalveolar lavage fluid was collected from the right middle lung lobe for the measurement of total protein (**c**) and for the quantification of macrophages (**d**) and neutrophils (**e**). Cumulative histopathology scores were obtained following blinded microscopic examination of fixed trachea (**f**) and left lung tissue (**g**). Individual data points (jittered horizontally to remove overlap), overlayed with model estimates (diamonds) and standard errors, are displayed for each experimental group, with the levels and difference compared to vehicle control listed underneath. Between-treatment comparisons to vehicle: * = *p* < 0.05, ** = *p* < 0.01, *** = *p* < 0.001, **** = *p* < 0.0001.

## Data Availability

The original contributions presented in the study are included in the article/Appendix A, further inquiries can be directed to the corresponding author/s.

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
