# Peer review of "Effect of TRPV4 Antagonist GSK2798745 on Chlorine Gas-Induced Acute Lung Injury in a Swine Model"

_ijms, 2024, doi:10.3390/ijms25073949_

Round 1

Reviewer 1 Report

Comments and Suggestions for Authors

The manuscript “TRPV4 Antagonism Results in Marginal Improvement of Chlorine Gas-Induced Acute Lung Injury in a Swine Model” by David J. Behm et colleagues focuses on providing evidence for the translatability of GSK2798745 as a new therapeutic measure for chlorine-induced ALI and ARDS. The authors based their studies on previously published literature suggesting that the TRPV4 inhibition may be beneficial in mice exposed to Chlorine. Their work on Yorkshire Swine is necessary for the further development and possible IND-enabling studies of this drug. The research is divided into a first dose-dependent study of Chlorine exposure in Swine and a second efficacy study of different doses of the TRPV4 inhibitor on a chosen dose of Chlorine. The results presented are negative, but this doesn’t necessarily mean that the manuscript needs to be rejected. However, the authors need to focus on a clear presentation of the results, identification of key negative elements during their research, and a better discussion with limitations and future directions. Indeed, the manuscript even if presenting negative data, may help other researchers in not pursuing these studies or applying important changes to the study protocols and methods. Major comments below:

 1.    Study design: There are only 2 animals for the control group. This is insufficient for a proper representation and statistical analysis. Authors need at least 1-2 additional animals for the control group.

2.    Results: The result section is confusionary. The authors use continuous monitoring, fitting curves, and a mathematical approach to biological research. As a result, a real dose dependency, statistical difference between groups, and efficacy of the inhibitor are not identified properly. While this approach may be useful for a drug that shows a clear benefit, it doesn’t help here.

                   i.    Section 2.1 should have a clear description of the groups and a logical progression of the results presented below. Authors are encouraged to rewrite the whole section. The figure 1 needs to be redone. The legend is written in small fonts and is hard to read. The statistical difference should be clear from looking at the images and by reading the image legend. Authors can keep the continuous line graphs as supplementary materials but need bar graphs to identify the statistical difference between groups at chosen time points post-chlorine exposure.

                                        ii.    Figure 2. The dose dependency is not clear. While there is a visible difference between groups, it is not clear if there is an actual statistical difference among animals exposed to different doses for both Ppeak and Compliance. Authors are either encouraged to use bar graphs with individual values at chosen time points or perform a 2-way ANOVA with Bonferroni post-test for the actual continuous lines graph.

                                       iii.    Section 2.2/Figure 3: Figures 3C and 3D do not appropriately represent the data collected. The use of a four-parameter fitting curve is not appropriate for this data. Use bar graphs to represent the data and perform statistical analysis with 1-way ANOVA and appropriate post-tests.

                                       iv.    Figure 4 lacks the histological pictures from control animals.

                                        v.    Section 2.3. The authors plot the AUC for continuously evaluated P/F ratio between groups. This is not an appropriate analysis, with the risk of creating further confusion, limited reproducibility for other authors, and misinterpretation of the actual results. The data needs to be shown clearly at different time points. This will help in 1) clearly show if there is a difference in P/F ratio at the end of the experimental period 24 hours, and 2) show if there is a temporary improvement in P/F ratio. This may help in identifying new dosage protocols or new strategies to improve the profile of activity of this drug.

                                       vi.    Figure S4 is confusionary. It shows almost no variation after intravenous infusion of the TRPV4 inhibitor, while the half-life of the drug is 13h. Please use a non-logaritmic y-scale as done by others (https://link.springer.com/article/10.1007/s40256-018-00320-6).

Discussion: The authors need to improve the discussion with the limitations of the study and further speculate on the lack of efficacy of the drug.: e.g. translatability of the drug in a swine model, wrong dosage protocols, binding of the Swine TRPV4, concentration OF GSK-drug in the lung alveolar space.

Title: The only section of the manuscript that as of now shows an “improvement” in section 2.3. The authors are encouraged to replot the data following the comments and if there is no positive data, to change the title accordingly: e.g. “Inhibition of TRPV4 does not improve Chlorine-induce ARDS at 24hours”.

Comments on the Quality of English Language

The English style is very direct, repetitive and lacks appropriate pausing and connections between phrases. As a result, the article is difficult to read. This in addition to the lack of clear figures and legends, and lack of beneficial results,  makes the article unpleasant to the reader.

Reviewer 2 Report

Comments and Suggestions for Authors

Authors proposed a paper entitled: “TRPV4 Antagonism Results in Marginal Improvement of Chlorine Gas-Induced Acute Lung Injury in a Swine Model” for the publication in IJMS, MDPI.

The paper has a good scientific soundness, but it should be improved, especially in terms of state of the art description. Please improve the introduction with the addition of more literature references.

In general, I believe that the caption of figures could be significantly reduced, maybe incorporating most of their length in the manuscript text description. However, this should depend on the authors’ style.

I suggest adding an abbreviation list, according to the guidelines of this Journal.

Here is the list of my other issues:

I am not sure that “GSK2798745” could be a keyword.

Line 30. I would suggest starting in this manner the introduction: “Chlorine stands as one of the most extensively manufactured and utilized chemicals worldwide. Accidental encounters with chlorine gas are common occurrences in both occupational and domestic environments. Moreover, documented instances reveal substantial public exposures resulting from auto/rail accidents linked to the transportation of chlorine.”

As an overall comment, the introduction provides a comprehensive overview of the significance of chlorine exposure and its associated health impacts. However, to enhance clarity and completeness, you may consider explicitly mentioning the purpose and objective of the study or investigation. Here's a suggested addition:

"While the introduction highlights the widespread use of chlorine and its potential health hazards, it is crucial to explicitly state the study's purpose and objective. For instance, you may include information about the specific focus on evaluating the effectiveness of GSK2798745 as a post-exposure therapeutic in a mechanically ventilated Yorkshire swine model of chlorine gas-induced acute lung injury. This addition will provide readers with a clear understanding of the research goal and the context in which the study is conducted."

The introduction needs to be expanded properly. Additionally, at the end of the introduction, a paragraph could state clearly what are the goals and the intentions of the authors with this paper.

I suggest improving the focus of figures 1.

Line 105. “This was followed by a more gradual increase in re-105 sistance from 30 minutes through the end of exposure (Figure S2a)” why the choice of not inserting this data instead of separating it in the supplementary data?

Line 138. “At exposure times of 55 minutes or longer, however, PC200 values were not correlated with the chlorine dose.” I would correct in this manner. Additionally, “minutes” could become “min”.

Figure 3 needs to be improved in terms of focus.

Line 170. This passage is correct. It effectively describes the histological findings in chlorine-exposed animals, detailing evidence of airway injury and inflammation distributed throughout the respiratory tract. The description is clear and concise, providing specific information about the nature and location of the observed pathology.

However, author could consider the possibility to re-write it in this manner:

“Chlorine-exposed animals showcased indications of airway injury and inflammation, distributed in a gradient across the respiratory tract. The trachea and proximal bronchi exhibited the most severe pathology, featuring extensive ulceration, fibrin accumulation, neutrophilic inflammation, reduced bronchial epithelial cell presence, and the absence of cilia. In the lower airways, intermittent sloughed epithelia formed sheets, blocking smaller bronchioles and alveolar ducts, whereas the alveolar epithelia remained largely unaffected (refer to Figure 4). These observed changes in tissue structure closely parallel the outcomes documented in swine subjected to 100-140 ppm chlorine through mechanical ventilation [19].”

Figure 4. The reference bar is not clearly visible.

Line 325: I suggest re-writing it as follows: “A significant challenge in this study arose from the necessity to choose a specific chlorine dose and dosing protocol to serve as a representative disease model for assessing the efficacy of GSK2798745. This limitation stems from the inherent variability in the disease phenotype resulting from chlorine exposure in humans, which is influenced by factors such as dose, exposure duration, and underlying health conditions.”

Section Nr. 4. Are there any references that could be added to these preparation methods descriptions?

Section 4.6. Please write formula according to the guidelines of this Journal. Same observation at the end of section 4.8.

I would dedicate a specific section at the end of the paper entitled “conclusion” including a critical summary of results, together with a paragraph reporting future perspectives.

Comments on the Quality of English Language

Moderate english revision needed. However, I suggested how to modify some paragraphs or sentences.

Reviewer 3 Report

Comments and Suggestions for Authors

The manuscript “TRPV4 Antagonism Results in Marginal Improvement of Chlorine Gas-Induced Acute Lung Injury in a Swine Model” is a very well written, carefully performed study. The aims of this study were first to establish a porcine model of lung injury after chlorine gas exposure and second to evaluate the dose-dependent effects of the TRPV4 inhibitor GSK2798745 on the lung injury induced by chlorine gas. The effects of exposure to chlorine gas for different amounts of time were evaluated by lung function, histology and cytology. A dose-response study of the TRPV4 inhibitor GSK2798745 was performed and marginal improvement of oxygenation was observed, but no effects on infiltration by inflammatory cells or histopathology were observed after treatment with GSK2798745. The references are relevant and the statistical analysis are correctly applied. However, there are some concerns with the study:

Major concerns:

1: Some of the authors should declare a conflict of interest as they are employees of GlaxoSmithKline, the company that developed GSK2798745.

2: Materials and methods: Please describe the cytology experiment. How was in performed and how were the cells stained? How were the cells counted?

Minor concerns:

Line 83: Do you mean oxygenation index?

Figure 4: Please include size of scale bars in the figure legend.

Author Response

The Authors thank Reviewer 3 for their diligent evaluation of our manuscript and detailed feedback provided.  The Authors have carefully considered each point and have made numerous modifications which we believe greatly improve the quality of the manuscript.  Specific responses to each point can be found below.

The manuscript “TRPV4 Antagonism Results in Marginal Improvement of Chlorine Gas-Induced Acute Lung Injury in a Swine Model” is a very well written, carefully performed study. The aims of this study were first to establish a porcine model of lung injury after chlorine gas exposure and second to evaluate the dose-dependent effects of the TRPV4 inhibitor GSK2798745 on the lung injury induced by chlorine gas. The effects of exposure to chlorine gas for different amounts of time were evaluated by lung function, histology and cytology. A dose-response study of the TRPV4 inhibitor GSK2798745 was performed and marginal improvement of oxygenation was observed, but no effects on infiltration by inflammatory cells or histopathology were observed after treatment with GSK2798745. The references are relevant and the statistical analysis are correctly applied. However, there are some concerns with the study:

Major concerns:

1: Some of the authors should declare a conflict of interest as they are employees of GlaxoSmithKline, the company that developed GSK2798745.

The following Conflict of Interest statement has been added for GSK authors: “Mathieu Bray, Mark Burgert, Scott Sucoloski, and David Behm were employees of GSK, the company that developed GSK2798745.”

2: Materials and methods: Please describe the cytology experiment. How was in performed and how were the cells stained? How were the cells counted?

Details of the cytology materials and methods have been added to the manuscript (see 5.11. Cytology).

Minor concerns:

Line 83: Do you mean oxygenation index?

Oxygen index has been changed to oxygenation index throughout the manuscript.

Figure 4: Please include size of scale bars in the figure legend.

This action has been addressed in response to Reviewer 2.

Round 2

Reviewer 2 Report

Comments and Suggestions for Authors

Authors provided a revised version of their paper.

Author correctly expanded the introductive sections, as also requested by the reviewers.

Line 92. “cmH2O/L/sec” after “cm”, this should be written as subscript.

Focus of figure improved.

Figure 3. I would increase the y scale axis, in order to better show the limit red dashed line.

Author Response

The Authors thank Reviewer 2 for their rapid evaluation of our manuscript re-submission and the additional feedback provided.  Our responses to each point, including specification of changes to the manuscript, can be found below.

Authors provided a revised version of their paper.

Author correctly expanded the introductive sections, as also requested by the reviewers.

Line 92. “cmH2O/L/sec” after “cm”, this should be written as subscript.

The authors respectfully disagree with Reviewer 2. The numerator unit is ‘centimeters of water’ and is therefore appropriately abbreviated as ‘cmH2O’. In support of this position, our use of the ‘cmH2O/L/sec’ abbreviation is consistent with the scientific literature (e.g., Hirayama et al, Acta Med Okayama, 2014, 68: 323-329; Erram et al, PLoS One, 2021, 16: e0252916).

Focus of figure improved.

Figure 3. I would increase the y scale axis, in order to better show the limit red dashed line.

Figure 3 has been updated as requested.